# Extracellular Vesicles in Advanced Prostate Cancer: Tools to Predict and Thwart Therapeutic Resistance

**DOI:** 10.3390/cancers13153791

**Published:** 2021-07-28

**Authors:** Carolina Saldana, Amene Majidipur, Emma Beaumont, Eric Huet, Alexandre de la Taille, Francis Vacherot, Virginie Firlej, Damien Destouches

**Affiliations:** 1Univ Paris Est Creteil, TRePCa, F-94010 Creteil, France; carolina.saldana@aphp.fr (C.S.); ami.majidi14@gmail.com (A.M.); emma.beaumont@orange.fr (E.B.); huet@u-pec.fr (E.H.); alexandredelataille@gmail.com (A.d.l.T.); vacherot@u-pec.fr (F.V.); virginie.firlej@u-pec.fr (V.F.); 2AP-HP, Hopital Henri-Mondor, Service Oncologie, F-94010 Creteil, France; 3AP-HP, Hopital Henri-Mondor, Service Urologie, F-94010 Creteil, France

**Keywords:** prostate cancer, extracellular vesicles, therapeutic resistances, predictive biomarkers

## Abstract

**Simple Summary:**

Advanced prostate cancer remains currently an important cause of cancer death. Despite the emergence of new treatments during the last decade, their efficiency is limited due to therapeutic resistance of tumor cells. Extracellular vesicles are secreted by cells and play an important role in cell–cell communications. Their content is specific to the cell that produced them, and they can be isolated from biological fluids such as blood and urine. In this review, we highlight the recent results demonstrating the impact of extracellular vesicles in the mechanisms leading to therapeutic resistance and their use to find new predictive biomarkers in order to facilitate treatment decision and personalized medicine.

**Abstract:**

Prostate cancer (PCa) is the second most frequent cancer and the fifth leading cause of cancer death among men worldwide. At first, advanced PCa is treated by androgen deprivation therapy with a good initial response. Nevertheless, recurrences occur, leading to Castrate-Resistance Prostate Cancer (CRPC). During the last decade, new therapies based on inhibition of the androgen receptor pathway or taxane chemotherapies have been used to treat CRPC patients leading to an increase in overall survival, but the occurrence of resistances limits their benefits. Numerous studies have demonstrated the implication of extracellular vesicles (EVs) in different cancer cellular mechanisms. Thus, the possibility to isolate and explore EVs produced by tumor cells in plasma/sera represents an important opportunity for the deciphering of those mechanisms and the discovery of biomarkers. Herein, we summarized the role of EVs in therapeutic resistance of advanced prostate cancer and their use to find biomarkers able to predict these resistances.

## 1. Introduction

In 2018, prostate cancer (PCa) incidence worldwide was estimated at 1.3 million new cases, associated with 359,000 deaths, ranking PCa as the second most frequent cancer and the fifth leading cause of cancer death among men [1].

Therapeutic approaches and prognosis of PCa depend on its stage at diagnosis (Figure 1). In 70% of cases, the diagnosis occurs when the tumor is localized in the prostate gland and 20% are diagnosed at the locally advance stage. Treatment of these tumors by surgery or radiotherapy guarantees a high survival rate. Nevertheless, despite the best standard of care treatment, one third of localized/locally advanced prostate cancer will relapse. In addition, in 5 to 10% of PCa, diagnosis occurs at the metastatic stage.

At a metastatic stage or after a progression from a localized tumor to a metastatic stage, patients enter into the advanced PCa phase (Figure 1). Androgen-deprivation therapy (ADT; i.e., LHRH antagonists or agonists) is the gold standard treatment, since prostate tumor cells are dependent on androgens to survive and grow at this stage [2]. Improved outcomes were observed by several practice-changing trials in an advanced setting such as the addition of androgen receptor pathway inhibitors (ARPIs) or docetaxel-based chemotherapy in combination with ADT [3,4,5]. However, during ADT, metastatic castrate-resistant prostate cancer (mCRPC) irremediably emerges because tumor cells adapt in order to survive and grow (PSA or radiological progression) despite a low blood level of testosterone [6]. Since 2004, first-line chemotherapy for the treatment of mCRPC essentially includes the docetaxel taxane, that inhibits microtubular depolymerization, in combination with ADT [7,8]. Since 2015, this therapy is also used for ADT untreated metastatic cancer, named metastatic hormonal sensitive or hormonal naïve cancer (mHSPC), improving progression-free survival (PFS) and an overall survival (OS) [3,4,9]. Additionally, Cabazitaxel, a second-generation semisynthetic taxane, is currently restricted for treatment of men with mCRPC with a disease which has progressed during or after docetaxel-based therapy [10]. Cabazitaxel failed to show survival benefit in the first-line setting compared to docetaxel [11].

Among androgen receptor pathway inhibitors (ARPIs), two groups can be described: those suppressing androgen biosynthesis (abiraterone) [12] and those blocking AR (enzalutamide, apalutamide and darolutamide) [13]. Abiraterone acetate and enzalutamide have demonstrated significant benefits in terms of OS and PFS either for chemo-naïve and locally advanced CRPC, mHSPC or in a post-docetaxel setting [14,15,16,17,18,19,20,21]. Concerning Apalutamide, a recent phase III trial led to its approval in mHSPC [22]. Finally, radionuclides such as radium 223 showed improved survival rates with minor bone marrow damage in mCRPC with bone-only metastasis [23].

Metastatic CRPC patients were expected to survive from 16 to 18 months [24]. Despite the emergence of these new drugs, the increased median survivals due to these new therapies ranged only from 1.9 to 8.2 months [7,8,10,14,15,18,20]. A part of the patients presents primary resistance: one third are refractory to docetaxel or abiraterone and one fourth to enzalutamide [8,14,18]. Furthermore, mCRPC tumor cells acquire additional resistances leading to patient death. Therefore, understanding the mechanisms of treatment resistance or sensitivity can help to achieve a more effective management of mCRPC to ensure better outcomes by limiting side effects and costs of unnecessary treatments or procedures. In addition, it could lead to the identification of potential therapeutic targets to prevent the emergence of such resistance.

Four main challenges must be solved in PCa (Figure 1). First, some patients are still diagnosed at the metastatic stage. New tools are needed to allow earlier detection. Secondly, as an indolent form of PCa is known to be livable without treatment and treating it could cause unnecessary side effects, indolent and aggressive PCa need to be discriminated. This makes biomarkers research an important axis in PCa. Additionally, to avoid unnecessary side effect of chemotherapy or ARPIs, identification of biomarkers predictive of therapeutic response remains very important to help practitioners in therapeutic decisions. Finally, as therapeutic resistances occur in advanced PCa, the understanding of resistance mechanisms should allow one to identify new molecular targets and define new therapeutic strategies to thwart them.

In this context, extracellular vesicles (EVs) may appear as very promising tools. Indeed, most cell types, including cancer cells, secrete EVs. They contain molecules such as DNA, RNA, proteins, lipids or metabolites reflecting the content of the cell from which they are derived. Furthermore, in pluricellular organisms, EVs can be isolated from different body fluids including blood, plasma/serum and urine, making them a powerful and accessible source of biomarkers [25].

In this review, we highlight the roles of EVs in therapeutic resistance mechanisms in an advanced PCa context and their promising use to discover new predictive biomarkers.

## 2. Extracellular Vesicles: Generalities

The concept of EVs has been first reported in 1946 as “procoagulant platelet-derived particles” and later as “platelet dust” circulating in the plasma [26,27]. Exosomes were then described in the 1980′s as EVs involved in the transport of the transferrin receptor out of reticulocytes during their maturation [28,29,30]. They were considered as artifacts or a trash bin for unnecessary and redundant proteins [31,32]. The International Society of Extracellular Vesicles currently defines EVs as “particles naturally released from the cell that are delimited by a lipid bilayer and cannot replicate, i.e., do not contain a functional nucleus” [33].

EVs represent a large and heterogeneous group of vesicles referenced under different names such as exosomes, ectosomes, microvesicles, oncosomes, prostasomes or exosome-like vesicles, depending on their tissue/cell of origin, nature, size or proposed function [34]. Based on the current knowledge of their biogenesis, EVs can be divided into two main groups: ectosomes/microvesicles and exosomes [35]. Ectosomes/microvesicles are produced by shedding from the plasma membrane. The activation of the plasma membrane is associated with intracellular calcium influx, mediated by ARF-6 and interactions between cytoskeletal resident proteins actin and myosin [36]. In general, they display a range of size from 50 nm to 1 µm. A subgroup of microparticules produced by tumor cells is called large oncosomes (LO). They were first described in prostate cancer, and they display an important size from 1 to 10 µm [37,38]. LO have been detected in tumor tissues and plasma from PCa patients and can contain DNA fragments [38].

Exosomes have an endosomal origin and are produced during maturation of multi-vesicular endosomes (MVEs) through intracellular budding of late endosomes generating intra-luminal vesicles (ILVs) [34,39]. Most of the MVEs are degraded after fusion with lysosomes but some of them fuse with the plasma membrane to release their content, the exosomes, in the extracellular compartment. The biogenesis of exosomes remains complex and depends on cell types and cargo transports [35]. Exosomes can be produced through the Endosomal Sorting Complex Required for Transport (ESCRT) machinery or through an ESCRT-independent pathway with the lipid ceramides or proteins from the tetraspanin family [40,41].

Since no consensus has been established concerning specific markers for determining the origin of EVs (exosome or ectosome), the International Society of Extracellular Vesicles (ISEV) advises now to use the generic term of EVs instead of exosomes or ectosomes. This generic term of EVs should be accompanied with an operational term referring to their size (small, medium, large) or density (low, medium or high) characteristics, composition (CD63+, CD9+, CD81+, Annexin A5, …) or origin (large oncosomes, apoptotic bodies, …) [33].

Methods to isolate EVs are based on their biophysical and biochemical properties. Ultrafiltration, differential centrifugation, density gradient centrifugation and size exclusion chromatography allow EV isolation due to their size and their density properties. Different EV purification kits are often polyethylene glycol based leading to EV precipitation in combination with low speed centrifugation. Other methods take advantage of EV antigen presence for their immunocapture with specific antibodies [42]. Currently, there is no standard method to isolate EVs without contaminants including soluble proteins, protein aggregates, lipoproteins and other particles such as viruses or organelles. The choice of the method should take into account the balance between the desired purity and concentration of EVs. Isolated EVs have to be characterized to determine their size, concentration, content and the presence of contaminants as recommended by the ISEV [33].

EVs are now considered as part of cell–cell communication system. Indeed, EVs allow the transfer of molecules such as DNA fragments, RNA, proteins, lipids or metabolites from an emitting cell to a recipient one. Furthermore, EVs appear as a promising biomarkers source since they can be isolated from different bodily fluids in pluricellular organisms including blood and plasma/serum [25,43,44], urine [45], saliva [46], breast milk [47], amniotic fluid, ascites, cerebrospinal fluid, bile, nasal secretions [48], feces [49] and semen [50].

Production of EVs from normal prostate cells has been described in the late 1970s. These EVs were called prostasomes and their origin (exosomes or microvesicles) is still a matter of debate [51]. Prostasomes display a range of size from 50 to 500 nm with a mean about 150 nm [52]. They can be isolated from seminal plasma [53] and urine [54] and can be produced by prostate cancer cells [55].

Accumulating evidence from the last decade, publications suggests a significant role of EVs in the hallmarks of cancer [56]. EVs were shown to be able to induce PCa cell proliferation [57,58,59], to promote the metastasis process through the acquisition of migration and invasion properties [60,61,62] and to block the immune system [63].

## 3. Role of EVs in Therapeutic Resistance in Advanced Prostate Cancer

### 3.1. Modulation of the AR Pathway

The androgen receptor (AR) axis plays a key role in the development of mCRPC. Most of the known mechanisms are based on the reactivation AR pathway despite a low level of blood testosterone, including AR gene mutation, emergence of AR splice variants, amplification and overexpression of the AR protein or intraprostatic production of androgens by tumor cells themselves [64]. Other mechanisms described are based on crosstalk with alternative signaling pathways [65].

Recent studies have suggested that EVs from tumor cells may transport AR and its variants to recipient cells [57]. These EVs containing AR were able to activate the expression of AR-responsive genes, as AR from EVs can bind to the enhancer region of PSA and increase RNA polymerase II occupancy. Therefore, these EVs promoted proliferation of LNCaP cells in the absence of androgen. A link between EVs and ADT resistance was proposed by Soekmadji et al. EVs were isolated from androgen-dependent LNCaP cells that were deprived of androgen or treated with the AR inhibitor enzalutamide. Proteomic analysis of these EVs highlighted the crosstalk between AR signaling, EV secretion pathways and calcium homeostasis, participating in the enzalutamide resistance [66]. Recently, Zhang et al. have demonstrated that LNCaP cells treated with EVs from androgen-independent cells (PC3) were able to acquire androgen-independent features with increased proliferation observed in both androgen-supplied and androgen-deprived medium [67]. Increased tumor growth of such treated cells was confirmed *in vivo* in intact and castrated NOD/SCID mice. The acquisition of the androgen-independent state was associated with a decrease in AR and PSA expression and an up-regulation of HMOX1 [68]. This up-regulation of HMOX1 was reported to be responsible for the acquisition of the androgen-independent state [67]. These studies suggest a role of EVs in the emergence of the mCRPC state after ADT (Figure 2); however, the mechanisms remain poorly investigated.

One of the mechanism responsible for ADT and ARPIs resistance is the emergence of AR splicing variants (AR-Vs). Endogenous expression of AR-Vs in PCa cell lines was first reported in the early 2000s [69]. Currently, more than 20 AR-Vs have been described and most of them exhibit a truncated COOH-terminal ligand-binding domain (LBD) with a remaining ability to bind DNA leading to a constitutive activation despite the absence of ligands [70]. AR splice variant-7 (AR-V7) is the most investigated variant accountable for ADT and ARPIs resistance of PCa tumor cells [71].

Full-length AR and its AR-V7 variant proteins were recently detected in EVs isolated from tumor prostate cell lines LNCaP and 22RV1 [57]. The treatment of AR negative PC3 cells with EVs isolated from AR positive 22RV1 cells led to a binding of AR to its promotor region and to the transcription activation of AR responsive genes demonstrating the AR transfer via EVs. Moreover, transfer of AR and AR-V7 to androgen-responsive cells promoted cell proliferation [57]. Presence of full-length AR and AR-V7 was confirmed in EVs from plasma of PCa patients [72,73,74].

These results highlight the role of EVs in the transfer of ARPIs resistance through AR and its AR-V7 variant (Figure 2).

### 3.2. EVs and Chemotherapy Resistance Transfer

The role of EVs in cancer resistance to chemotherapy was first suggested with the observation of a positive correlation between expression of membrane shedding-related genes by tumor cells and their profile of chemotherapy resistance [75]. The anticancer drug doxorubicin was found to be accumulated in the EVs of different cell lines, including PC3, suggesting a role of drug efflux leading to drug resistance. Based on this, inhibition of EVs production was associated with increased sensitivity to chemotherapy. Indeed, inhibition of EVs production with EV biogenesis inhibitor (chloramine and bisindolylmaleimide-I) led to increased concentration of 5-Fluorouracile (5-FU) inside PC3 cells, associated with increased apoptosis and inhibition of 5-FU resistance [76]. Inhibition of EVs production using the calpain inhibitors (Calpeptin) and siRNA, showed higher intracellular retention of docetaxel associated with a decrease in its GI_50_ and reduced tumor volume in PC3 ectopic xenograft in mice [77].

The most investigated mechanism to explain resistance to chemotherapy remains the overexpression of multidrug resistance (MDR) genes, also called ATP-binding cassette (ABC) transporter genes. The transporter proteins encoded by these genes play the role of a molecular pump leading to a decrease in the intracellular concentration of drugs. Several MDR proteins such as ABCB1 (or MDR1/P-glycoprotein), ABCC4 or ABCC5 are overexpressed in PCa cells, contributing to taxane resistance [78,79]. Lucotti et al. have demonstrated that treatment of DU145 cells with fludarabin decreased hsa-miR-485-3p in EVs with its cell retention leading to inhibition of the transcription factor NF-YB associated with an increase in MDR-1 expression and fludarabin resistance [80].

To investigate the role of EVs in chemoresistance, tumor prostate cell lines were treated with chemotherapy agents until the acquisition of resistance. Several studies followed the strategy consisting of EVs isolation from the different cell lines, and treatment and observation of sensitivity/resistance to chemotherapy transfer between cells. Via this strategy, EVs protein content was compared between docetaxel sensitive and resistant DU145 prostate tumor cell lines. Among the protein differentially present in the EVs from the two different cell lines, MDR-1 and MDR-3 proteins, involved in drug efflux, were found enriched in the EVs of docetaxel-resistant cells [81]. Corcoran et al. used EVs isolated from docetaxel-resistant 22RV1 and DU145 cell line variants. Treatment of the parental 22RV1 and DU145 cell lines with these EVs did not modify cell proliferation, migration or invasion but was able to confer to them resistance to docetaxel. Furthermore, EVs isolated from sera of patients undergoing docetaxel treatment were also able to transfer docetaxel resistance to DU145 and 22RV1 cells when patients displayed docetaxel resistance. One mechanism of docetaxel resistance was the transfer of MDR-1/P-glycoprotein via the EVs [82].

A similar study has demonstrated that camptothecin-sensitive DU145 cells treated with EVs isolated from camptothecin-resistant DU145 cells acquired a higher resistance to camptothecin, observed by a resistance to apoptosis linked to reduced cleaved PARP [83]. In the inverse strategy, EVs were isolated from non-malignant prostate cells and human mesenchymal stem cells (RWPE-1 and MSCs) and used to treat a paclitaxel resistant DU145 cell line. EVs from both non-malignant cell lines were able to reverse the resistant phenotype of paclitaxel resistant DU145 cells: those cells subsequently encountered apoptosis when treated by EVs [84]. Using two prostate tumor cell lines resistant to enzalutamide, C4-2B and CWR-R1, it was demonstrated that resistant cells produced more EVs than their parental counterparts. Mechanistic studies indicated that syntaxin was over-expressed in resistant cells compared to the sensitive one and could be responsible for the increase in EVs production [85]. EVs from PC3 and DU145 docetaxel resistant cells were isolated and their content in miRNA was analyzed using the miRNA microarray chip. Bioinformatic analysis permitted one to identify the miRNAs able to regulate AR, PTEN and TCF4 genes and to confer resistance to docetaxel. miR3176, -141-3p, -5004-5p, -16-5p, -3915, -488-3p, -23c, -3673 and -3654 regulated AR and PTEN whereas TCF4 is a target of miR-32-5, -141-3p, -606, -381 and -429 [86].

All these results determine that EVs are able to transfer resistance information to therapeutic sensitive cells (Figure 2). The understanding of involved mechanisms is critical to develop strategies to block these cell-to-cell communications.

### 3.3. Lineage Plasticity (NED and EMT)

One of the therapeutic resistance mechanisms described in mCRPC is the emergence of a high rate of tumor cell heterogeneity. Androgen deprivation has been shown to activate both epithelial-to-mesenchymal transition (EMT) and neuroendocrine trans-differentiation (NED) programs.

EMT is well known for the promotion of biological phenotypes associated with tumor progression (migration/invasion, tumor cell survival, cancer stem cell-like properties and resistance to radiation and chemotherapy) in multiple human cancer types [87].

In previous studies, we have generated a prostate tumor cell line derived from the 22RV1 cell line and named Mes-22RV1 [88]. This new cell line had undergone the EMT process and displayed mesenchymal features. These Mes-22RV1 cells produce more EVs compared to 22RV1 parental cells. The role of EVs isolated from these mesenchymal prostate tumor cells in the induction of EMT in epithelial VCaP cells was then investigated [89]. Recipient epithelial VCaP cells treated with EVs from the mesenchymal cells gained mesenchymal features accompanied with inhibition of AR expression and activity, increase in EMT markers (SNAI1, SNAI2, VIM, CDH2, PAI1 and FN1) and higher migration and invasion capacities. All these effects were associated with an increased resistance to enzalutamide [89]. Recently, it has been shown that EVs can deliver caveolin-1 into PCa cells leading to the cancer stem cell (CSC) phenotype. In LNCaP cells, caveolin-1 induces EMT through activation of the NFKB signaling pathway. This process, due to EMT induction, was associated with increased expression of ZEB1, ZEB2, Slug, Twist, vimentin and downregulated E-cadherin expression. Furthermore, LNCaP cells treated with caveolin-EVs exhibited an increase in migration and invasion capacities as well as higher resistance to docetaxel and radiotherapy. EMT induction was confirmed by overexpression of vimentin and downregulation of E-cadherin in Caveolin overexpressed LNCaP. Decreasing caveolin-1 expression using shRNA produced opposing results, confirming thus the role of caveolin-1 [90].

NED in prostate cancer is associated with resistance to therapy, visceral metastasis and aggressive disease. Thus, activation of such programs via the inhibition of the androgen axis provides a mechanism by which tumor cells can adapt to promote disease recurrence and progression [91,92,93].

The role of EVs in NED through the transfer of adipocyte differentiation-related protein (ADRP) was reported [94]. Inducing NED with IL-6 treatment or androgen deprivation led to increased expression of ADRP and packaging in EVs. ADRP overexpressing EVs triggered NED in C4-2 and DU145 cells. Furthermore, EVs isolated from C4-2B cells resistant to enzalutamide contain a higher amount of ADRP [94]. Recently, Bhagirath et al. have demonstrated the role of Pro-neural Pou-domain transcription factors, BRN4 and BRN2, in the induction of the neuroendocrine phenotype using IL-6 treatment [95]. During NED induction with IL-6 treatment, LNCaP cells produced more EVs. These EVs contained BRN4 and BRN2 mRNA. It has to be specified that these two mRNAs were also detected overexpressed in blood EVs from CRPC patients with neuroendocrine features. Enzalutamide treatment was also shown to enhance the release of BRN4 and BRN2 mRNA in PCa EVs, and treatment of LNCaP cells with these EVs promoted NED induction [95].

Accumulated treatment pressure in mCRPC patients increases the rate of tumor heterogeneity leading to highly resistant tumors [96]. These studies demonstrate the key role played by EVs in the establishment of tumor heterogeneity through EMT and NED leading to therapeutic resistance (Figure 2).

### 3.4. Role of the Microenvironment

The microenvironment of tumors is composed of different cell types such as cancer associated fibroblast (CAF), tumor associated macrophages (TAM), pericytes, immune and endothelial cells. All these cell types are known to be involved in therapeutic resistance [97].

Only a few studies have addressed the question of the role of EVs from the tumor microenvironment in PCa therapeutic resistance. Zhang et al. have studied the impact of EVs produced by primary CAF from PCa patients under ADT conditions on PCa tumor cells behavior. LNCaP and DU145 cells treated with these EVs displayed higher migration and invasion capacities associated with EMT process induction through a decrease in miR-146a-5p exosomal transfer from CAF to PCa cells under ADT. Confirming the *in vitro* results, the expression of miR-146a-5p was lower in the tumors of patients who received ADT compared to the tumors of patients who underwent surgery without ADT therapy [98]. Exosomes isolated from PSC-27cells (primary prostate fibroblasts) and added to PC3 cells induced chemotherapy resistance with increased IC_50_ values of 0.3 fold for cisplatin, 0.4 fold for doxycycline and 1.3 for docetaxel. This induced resistance was due to miR-27a derived exosomes that regulated the expression of Trp53 [99]. A third study showed that CAF-secreted exosomal miR-423-5p promoted resistance to docetaxel or bicalutamide in PCa cells LNCaP, 22RV1 and C4-2 cells, respectively, by targeting GREM2 [100].

## 4. EVs as Source of Biomarkers to Predict Therapeutic Resistances in PCa

### 4.1. EVs Content to Find New Predictive Biomarkers

EVs can be isolated from different biological fluids and their content reflects the one of the cell producing them. Consequently, they can be used as biomarkers sources for PCa diagnostic, prognostic and predictive response to treatments [52,101]. Concerning predictive response biomarkers, it is essential to note the difficulty of finding them within prostate tumor tissues since biopsies are not recommended in patients with advanced PCa. In this context, EVs isolated from liquid biopsies such as serum, plasma and urine, appear as an attractive source of material especially for prostate cancer personalized medicine. Over the past decade, urinary and blood based biomarkers from EVs have been described and used for diagnosis as well as for tumor progression prediction [102,103]. Among those biomarkers, some have been documented in relation to therapeutic resistances in PCa (Table 1).

By producing docetaxel resistant cells from prostate tumor cell lines, P-gp/MDR1 was demonstrated to be increased in EVs from docetaxel resistant prostate cells (PC-3, 22RV1, DU-145) as compared to those from parental cells [82,104]. These results were confirmed in blood EVs from docetaxel resistant patients as compared to blood EVs from therapy-naïve or sensitive patients [81,104]. In addition, the copy number of CD44v8-10 mRNA was higher in EVs isolated from the serum of docetaxel-resistant patients compared to those of docetaxel-naïve and control patients, whereas it was not observed for CD44 mRNA [105]. AR splice variants, especially AR-V7, are associated with the resistance to ARPIs and taxane based treatments. Presence of AR-V7 mRNA variant in EVs from PCa patients was demonstrated using plasma samples. AR-V7 mRNA in EVs was associated with faster resistance to gold standard therapies in CRPC patients [72,73]. Moreover, the presence of AR-V7 mRNA variant in EVs allowed one to distinguish responders from non-responders to diverse therapies such as abiraterone, enzalutamide, cabazitaxel and docetaxel [106]. In another study, the double positivity for AR gain function (based on cfDNA) and AR-V7 mRNA (ddPCR on EVs) was related to shorter PFS and overall survival on abiraterone and enzalutamide treated CRPC patients [107]. ITGB4 and vinculin have been outlined as upregulated in taxane resistant PC3 and could be potential biomarkers for aggressive tumors, although no clinical studies have been carried out on these markers [108]. TSP1 was revealed to be expressed in EVs derived from LNCaP-AR-Enzalutamide resistant cells and NCI-H660 cells in contrast to EVs from LNCaP-AR cells. This protein level was also increased in EVs purified from sera of CRPC patients with neuroendocrine features. These results suggest that TSP1 could be employed as a biomarker to predict resistance to ARPIs and to diagnose NED in CRPC patients [109]. Another study described the expression of YAP1 and COUP-TFII in EVs from LNCaP-Enzalutamide resistant cells as well as sera from patients resistant to enzalutamide [110].

Regarding resistance to radium-223, transcriptome analysis of EVs from treated patients revealed a change in bone-related pathways as well as DNA damage repair and immune checkpoint. Notably, PD-L1 expression is associated with a short survival to Radium-223 [111].

miRNAs could also be used as biomarkers for prostate cancer [101]. Nevertheless, only a few miRs were described as predictive biomarkers for therapies resistance in prostate cancer. miR-34a level was decreased in both cells and EVs of two docetaxel resistant cell lines (22Rv1RD and PC3RD). This miR-34a was significantly diminished in prostate cancer vs. normal tissue, in biochemical recurrence vs. non-recurrence tissue and in metastatic tissue vs. primary site. However, clinical resistance has not been explored [112]. A miR-array study has also identified 29 deregulated miRNAs, 19 upregulated and 10 downregulated, in EVs samples derived from two paclitaxel resistant PCa cells (PC3 and DU145) compared with their parental cells, but miR expression in patients has not been investigated [86]. miR-21 expression was increased in EVs from enzalutamide resistant LNCaP cells as well as in sera of enzalutamide resistant patients [110].

In conclusion, some biomarkers, listed in Table 1, have been reported in EVs from therapy resistant tumors and cells. However, nowadays, no useable signatures in the clinical test existed. Thus, the field remains open for research.

### 4.2. Specificities of Large Oncosomes

Recent developments in molecular biology techniques (digital droplet PCR, New Generation Sequencing) have rendered the accurate analysis of genetic alterations from tumor DNA (tDNA) possible. The amount of circulating tDNA (ctDNA) in plasma is proportional to the tumor mass and the overall shorter survival of CRPC patients [113]. The use of ctDNA as a non-invasive and specific blood biomarker could therefore represent an attractive approach to monitoring the evolution of the disease and to offering personalized treatments to CRPC patients. However, ctDNA in men with metastatic prostate cancer is rapidly reduced with ADT, thus limiting the detection of clinically relevant somatic mutations and subsequently compromising the use of ctDNA [114]. Furthermore, the half-life of ctDNA is relatively short (<2 h) since this DNA is subject to degradation by DNAses.

Recently, EVs have been shown to contain double stranded DNA carrying mutations identical to those found in parental tumor cells [115,116,117]. Certain genetic alterations such as TP53 or PTEN have thus been identified [118], suggesting that EVs contain genomic signatures characteristic of the cells from which they are derived. Moreover, it has been suggested that molecules present in EVs such as DNA and RNA are more stable as they are protected from enzymatic degradations [119].

A recent study on a small number of blood samples from CRPC patients has shown the possibility of isolating tDNA from large extracellular vesicles also called large oncosomes (LOs) [120]. Di Vizio’s group reported LOs to be specifically secreted by prostate tumor cells at a quantifiable rate [37]. Purified from tumor cells, they promote the migratory and invasive capacities of recipient cells [37,121]. LOs have been detected in tumor tissues and plasma of patients with metastatic PCa. Their number is directly correlated with aggressiveness of the disease [38,122].

LOs appear to be more promising for a future clinical use. With a diameter of 1 to 10 µm, they are 100 to 1000 times larger than exosomes and contain proteins, RNA and DNA [123]. Moreover, LOs have also been shown to be a subpopulation of EVs containing DNA with tumor-specific molecular content [122]. DNA purified from LOs issued from CRPC patient plasma is of high molecular weight (100 kbp–2 Mbp) and contains histones, hence suggesting the presence of intact chromosomes in these vesicles. This DNA appears to be specific since it is absent and not detected in the EVs of healthy subjects. NGS sequencing has shown that LOs DNA contains the same genetic alterations as those found in tumor tissue, such as variations in the copy number of genes frequently altered in CRPC (MYC, AKT1, PTK2, KLF10 and PTEN) [120].

A recent study described the possibility to purify circulating tumor cells and LOs in order to confer a higher sensitivity to the liquid biopsy assay in the case of CRPC. CTCs and LEVs could be discriminated based on size, morphology and DNA content [124].

### 4.3. Future of EVs Biomarkers

Several clinical studies using EVs content as biomarkers are conducted nowadays and, more particularly, on PCa diagnosis (exosome Dx; Sentinel™ PCC4 Assay, ClarityDX Prostate) [125]. Concerning therapeutic resistance, very few studies are in process. A clinical study is currently investigating the possible use of AR-V7 for predicting resistance under androgen-receptor signaling inhibitors (PEARL, NCT03601143).

If EV content appears as a promising source of biomarkers, their characterization and isolation methods need to be improved. Indeed, it remains challenging to isolate EVs from contaminants such as proteins without reducing the quantity of material needed for further analysis. In addition, gold standard methods have to be established in order to be applied in a clinical context [126]. Furthermore, it still remains difficult to discriminate between different EVs such as microvesicles and exosomes.

## 5. Conclusions

In advanced PCa, understanding the mechanisms of therapeutic resistance and finding new predictive biomarkers for therapeutic sensitivity are currently two of the main challenges for research. For years, the interest of EVs studied as tools to discover biomarkers and communication tools for cells has been growing. Their roles in the induction of therapeutic resistance in advanced prostate cancer began to be more detailed.

In the near future, EVs should allow a better understanding of the different mechanisms involved in therapeutic resistance. Moreover, their employment as tools for predictive biomarkers may allow personalized medicine in the field of many cancers, including prostate cancer.

## Figures and Tables

**Figure 1 cancers-13-03791-f001:**
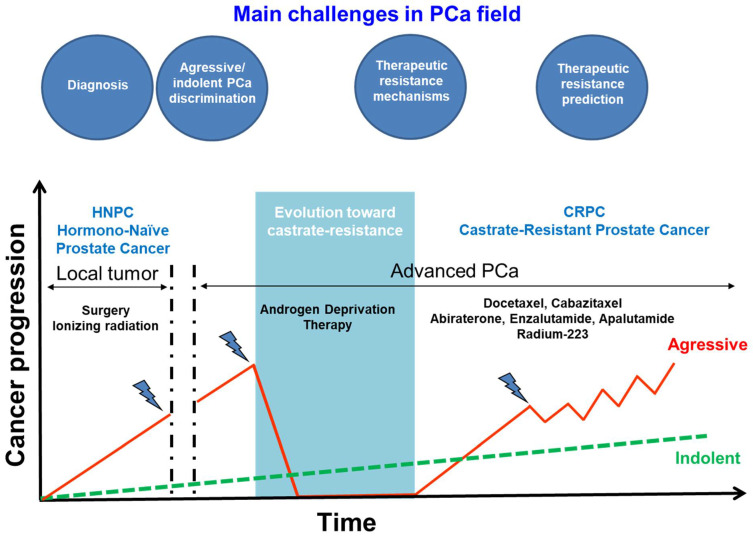
Clinical evolution of PCa and the main challenges for research.

**Figure 2 cancers-13-03791-f002:**
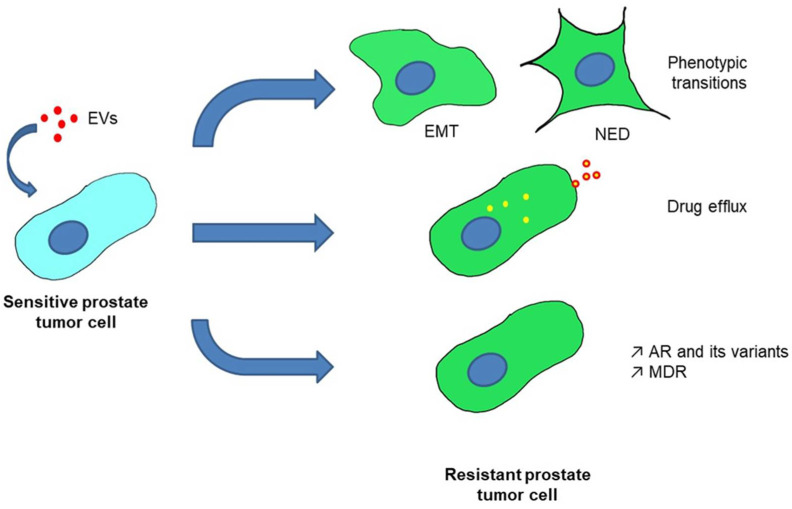
The main mechanisms of therapeutic resistance induced by EVs in prostate tumor cells. Roles of EVs have been demonstrated in the induction of tumor plasticity, elimination of drug inducing drug efflux or augmentation of gene expression or protein/mRNA transfer via EVs. EMT = Epithelial to Mesenchymal Transition, NED = Neuroendocine Differentiation, AR = Androgen Receptor and MDR = Multi Drug Resistance.

**Table 1 cancers-13-03791-t001:** List of biomarkers from EVs studied to be predictive of resistance/sensitivity to advanced PCa therapies. In bold miRNA were verified by RT-qPCR. Abbreviations: for EVs isolation, UC = Ultracentrifugation, SEC = Size Exclusion Chromatography, UF = Ultrafiltration; for analysis methods, ddPCR = droplet digital PCR, TLDA = Taqman Low Density Array.

Header Analyzed molecules	Up/Down in Resistant Cells or Tumors	Analysis Methods	EVs Isolation	Origin of Cells EVs	Origin of Patients EVs	Drug	Reference
**miRNA (bold: verified by RT-qPCR)**							
**hsa-miR-16-5p**, miR-203a, **miR-32-5p**, miR-515-3p, **miR-99b-5p**, **miR-451a**, **miR-1204**, miR-4291, miR-4291, miR-3673, **miR-23c**, miR-3654, **miR-3607-3p**, **miR-3915**, miR-4716-3p, miR-4722-5p, miR-488-3p, miR-4669, miR-5004-5p	Up	Microarray chip analysis validated by RT-qPCR	UC	DU145 and PC3		Paclitaxel	Li et al., 2016
**miR-141-3p**, **miR429**, miR192-5p, **miR192-3p**, miR-606, **miR-3176**, miR-1224-3p, miR-381-3p, miR-933, miR-34b-3p	Down	Microarray chip analysis validated by RT-qPCR	UC	DU145 and PC3		Paclitaxel	Li et al., 2016
miR-598, 148a, 34a, 146a	Down	TLDA with qPCR	UC	22RV1, DU145 and PC3	No for chemoresistance	Docetaxel	Corcoran et al. 2014
miR-21	Up	qPCR	SEC	LNCaP and LNCaP enz-resistant	Resistant vs. sensitive patients	Enzalutamide	Lee et al., 2021
**Protein**							
Pg-P	Up	Western-blot	UC	PC3	Resistant vs. therapy naive	Docetaxel	Kato et al., 2015
Pg-P	Up	Western-blot	UC	DU-145	Resistant versus sensitive	Docetaxel	Kharaziha et al., 2015
Pg-P	Up	Western-blot	UC	DU-145/22RV1		Docetaxel	Corcoran et al., 2012
Vinculine, ITGB4	Up	LC-MALDI-TOF	UC	PC3		Taxane (paclitaxel/docetaxel)	Kawakami et al., 2015
TSP1	Up	Western-blot	UF and precipitation	LNCaP-AR		Enzalutamide	Bhagirath et al., 2021
YAP1 COUP-TFII	Up	Western blot	SEC	LNCaP	Resistant vs. sensitive patients	Enzalutamide	Lee et al., 2021
PD-L1	Up	Western-blot, Luminex	UC		Favorable vs. unfavorable overall survival	Radium-223	Vardaki et al., 2021
**mRNA**							
CD44v8-10 mRNA	Up	RT-qPCR	UC		Resistant vs. naive	Docetaxel	Kato et al., 2020
AR-V7	presence	ddPCR	UC		Responders vs. non-responders	Abiraterone, enzalutamide, cabazitaxel, docetaxel	Foroni et al., 2020
AR-V7	presence	ddPCR	UC			Abiraterone, enzalutamide, docetaxel	Joncas et al., 2019
AR-V7	presence	ddPCR	Exoeasy spin columns			Enzalutamide, abiraterone	Del Re et al., 2017 Del Re et al., 2021
**Pathways modified**							
Bone related pathways DNA damage repair-related pathways Immune suppressors		RNAseq	UC		Favorable vs. unfavorable overall survival	Radium-223	Vardaki et al., 2021

## Data Availability

Not applicable.

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
