# Peer review of "Extracellular Vesicles in Advanced Prostate Cancer: Tools to Predict and Thwart Therapeutic Resistance"

_cancers, 2021, doi:10.3390/cancers13153791_

Round 1

Reviewer 1 Report

In this review, Saldana and colleagues reported on the possible use of EVs in advanced prostate cancer as tool to predict and thwart therapeutic resistance.

This review is quite interesting and provided an extensive number of published studies.

The manuscript is well written and comprehensive and English language and style are fine. However, please remove the refuse at page 4 after the citation n.61.

Please consider the suggestions below.

-It would be nice to include some notices on the use of EVs as biomarker in urine samples instead of plasma samples for prostate cancer in the session number 4.

For example, evaluate to include the papers mentioned below by Hendrix, Schalken and colleagues and even more.  

-Unravelling the proteomic landscape of extracellular vesicles in prostate cancer by density-based fractionation of urine

- Blood-based and urinary prostate cancer biomarkers: a review and comparison of novel biomarkers for detection and treatment decisions.

-In my opinion, there is a lack in describing the methods utilized for the isolation and the characterization of the extracellular vesicles. I think that in the session number 2 it would be interesting to introduce a short paragraph to refer to the EVs isolation and characterization if it is possible.

Minor suggestion:

-I’m not agree with the phrase at the end of page 9 in which the authors affirm that the volume of 8-10 mL of plasma is too much for further analysis on tDNA. I think that this volume doesn’t must be considered as a problematic point on this topic but on the contrary as a reasonable volume of blood necessary to implement the use of the analysis of tDNA in liquid biopsy. If you agree with this thought please think about rewriting this sentence.

Author Response

We would like to thank the reviewer for its comments and suggestions. As required, we have enclosed a detailed list of the changes, point by point.

- However, please remove the refuse at page 4 after the citation n.61.

This paragraph remained from the template. It has been removed.

-It would be nice to include some notices on the use of EVs as biomarker in urine samples instead of plasma samples for prostate cancer in the session number 4.

For example, evaluate to include the papers mentioned below by Hendrix, Schalken and colleagues and even more.  

-Unravelling the proteomic landscape of extracellular vesicles in prostate cancer by density-based fractionation of urine

- Blood-based and urinary prostate cancer biomarkers: a review and comparison of novel biomarkers for detection and treatment decisions.

Our review is focused on biomarkers described for treatment resistance of CRPC patients. To our knowledge, there are no articles describing urinary EVs used as biomarkers for therapeutic resistance. However, we add sentences concerning the use of urinary and blood EVs for diagnostic and prognostic of prostate cancer and include one paper suggested by the reviewer:

 “Over the past decade, urinary and blood based biomarkers from EVs have been described and used for diagnosis as well as for tumor progression prediction [102] [103]. Among those biomarkers, some have been documented in relation with therapeutic resistances in PCa (table 1)» (line 383-386)

-In my opinion, there is a lack in describing the methods utilized for the isolation and the characterization of the extracellular vesicles. I think that in the session number 2 it would be interesting to introduce a short paragraph to refer to the EVs isolation and characterization if it is possible.

Numerous reviews are specialized in the description of the EV isolation and characterization. Since there is a limited number of words allowed, we summarized this part as:

« Methods to isolate EVs are based on their biophysical and biochemical properties. Ultrafiltration, differencial centrifugation, density gradient centrifugation and size exclusion chromatography permit EV isolation due to their size and their density properties. Different EV purification kits are often polyethylene glycol based leading to EV precipitation in combination with low speed centrifugation. Other methods take advantage of EV antigen presence for their immunocapture with specific antibodies [42]. Currently, there is no standard method to isolate EVs without contaminants including soluble proteins, protein aggregates, lipoproteins and other particules such as viruses or organelles. The choice of the method should take in account the balance between the purity and the concentration of EVs desired. Isolated EVs have to be characterized to determine their size, concentration, content and the presence of contaminants as recommended by the ISEV [33]» (line 160-171)

Minor suggestion:

-I’m not agree with the phrase at the end of page 9 in which the authors affirm that the volume of 8-10 mL of plasma is too much for further analysis on tDNA. I think that this volume doesn’t must be considered as a problematic point on this topic but on the contrary as a reasonable volume of blood necessary to implement the use of the analysis of tDNA in liquid biopsy. If you agree with this thought please think about rewriting this sentence.

We would like to thank the reviewer for this comment. We change the sentence as:

« However, ctDNA in men with metastatic prostate cancer is rapidly reduced with ADT, thus limiting the detection of clinically relevant somatic mutations and subsequently compromising the useful of ctDNA (Vandekerkhove et al., 2019). Furthermore, the half-life of circulating tDNA is relatively short (<2 hours) since this DNA is subject to degradation by DNAses.”  (line 446-450)

Reviewer 2 Report

This review focused on the role of extracellular vesicles (EVs) and the possibility of using EVs as biomarkers in the therapeutic resistance prostate cancer (PCa). The review was very well written and covered the rational of looking for new biomarkers in PCa, the roles of EVs in the therapeutic resistance mechanism in advanced PCa, and the potential that EVs can be used as predictive biomarkers.

Here are some comments.

  1. In the introduction, the authors talked about the four main challenges in the PCa field. Could the authors please include more epidemiological information about incidence rates of each challenge? For example, the incidence of therapeutic resistance occurring in advance PCa and the related death, the percentage of indolent PCa and the incidence of severe side effects, etc. It will help the readers to understand better why these EV tools are important.
  2. Please delete the last paragraph (Line 152-155) of section 2.
  3. There are clinical trials exploring EVs for the diagnosis of prostate cancer. Please include these in the Section 4.
  4. Please also acknowledge the limitation of using EVs as biomarkers.

Author Response

We would like to thank the reviewer for its comments and suggestions. As required, we have enclosed a detailed list of the changes point by point.

  1. In the introduction, the authors talked about the four main challenges in the PCa field. Could the authors please include more epidemiological information about incidence rates of each challenge? For example, the incidence of therapeutic resistance occurring in advance PCa and the related death, the percentage of indolent PCa and the incidence of severe side effects, etc. It will help the readers to understand better why these EV tools are important.

Concerning the first challenge, to better diagnose PCa, the incidence of diagnosis at the metastatic stage (5 to 10%) is indicated (line 48).

Concerning the discrimination between the aggressive or indolent forms of PCa, the incidence of indolent form is difficult to estimate due to a lack of efficient markers. In reviews or articles, the indolent form is associated with the term “majority”. To avoid displaying wrong percentage, we prefer not to add estimation.

Concerning the therapeutic resistance, we add some incidence information:

« CRPC patients were expected to survive from 16 to 18 months [24]. Despite the emergence of several new efficient drugs, the increased median survivals due to these new therapies ranged only from 1.9 to 8.2 months [7,8,10,14,15,18,20]. A part of patients presents primary resistance: one third are refractoy to docetaxel or abiraterone and one fourth to enzalutamide [8,14,18]. Furthermore, mCRPC tumor cells acquire additional resistances leading to patient death.» (line 94-99)

  1. Please delete the last paragraph (Line 152-155) of section 2.

This paragraph remained from the template. It has been removed.

  1. There are clinical trials exploring EVs for the diagnosis of prostate cancer. Please include these in the Section 4.

Some clinical trials are in process. These promising trials have been included in a new section:

« 4.3. Future of EVs biomarkers

Several clinical studies using EVs content as biomarkers are conducted nowadays and, more particularly, on PCa diagnosis (exosome Dx; Sentinel™ PCC4 Assay, ClarityDX Prostate) [125]. Concerning therapeutic resistance, very few studies are in process. A clinical study currently investigates the possibile used of AR-V7 for predicting resistance under androgen-receptor signaling inhibitors (PEARL, NCT03601143 ).” (line 478-489)

  1. Please also acknowledge the limitation of using EVs as biomarkers.

The limitations of EVs as biomarkers have been added in the new section 4.3:

“If EV content appears a promising source of biomarkers, their characterization and isolation methods need to be improved. Indeed, it remains challenging to isolate EVs from contaminants such as proteins without reducing the quantity of material needed for further analysis. Also, gold standard methods have to be established in order to be applied in clinical context [126]. Furthermore, it remains still difficult to discriminate between different EVs such as microvesicles and exosomes.” (line 478-489)

Reviewer 3 Report

Extracellular vesicles in advanced prostate cancer: tools to predict and thwart therapeutic resistance

Interesting manuscript by Saldana et al. Authors nicely summarized and reported current knowledge on the roles of extracellular vesicles (EVs) in advanced prostate cancer as well as their potential for diagnosis purposes. Nevertheless, two major flaws greatly limit the overall quality of the present review: a lack of details and clear explanations when describing data from cited reports as well as poor writing quality. Please see details below.

  • Authors should provide further details about models (in vivo vs in vitro, cell line origins, experimental setups…etc) and describe cellular/molecular mechanisms implicated in the observed effects when reporting data from the cited publications. Authors need to report key experiments/data from cited publications and provide a clear description of the results. Often descriptions are vague and unclear: for instance, cf line 165-167; 169-170, line 237-238 – why partially? Please clarify -; line 245-246 – describe mechanisms if possible/available - ; line 252-253 – please expand, what miRNAs? - ; line 270-272 – please describe markers -; line 273-274 – clarify link between overexpression of caveolin-1 and EMT induction - ; line 312-313 – what’s the source of CAFs in the study? – (list is not exhaustive so please revise manuscript carefully).

  • The overall writing quality of the present paper is quite poor and need extensive revision. It makes reading complicated at times (cf line 24-25, ‘in prostate field’ line 44…etc). Please revise language / grammar with a native English speaker.

  • Line 150-152: a summary of the overall roles of EVs in PCa development and progression (not just resistance) would be a great addition to chapter 2. Try and describe some of the mechanisms mentioned in this paragraph (line 150-152). Also, general introduction to PCa and treatments could be shorter/more concise, so focus is more on the main subject here.

  • Line 152-155: please remove your own comments from pre-submission drafts…

  • Line 344-346: please clarify that the markers were found expressed in exosomes

  • Table 1 needs to be improved: it’s busy and layout is confusing; different colors / font sizes should be used to clearly discriminate specific sections

  • Further discussion on the potential of using EVs for therapeutic strategies would be great

  • Try and make conclusion more PCa-specific and expand on current challenges and future potential (clinical) applications

Author Response

We would like to thank the reviewer for its comments and suggestions. As required, we have enclosed a detailed list of the changes point by point.

  • Authors should provide further details about models (in vivo vs in vitro, cell line origins, experimental setups…etc) and describe cellular/molecular mechanisms implicated in the observed effects when reporting data from the cited publications. Authors need to report key experiments/data from cited publications and provide a clear description of the results. Often descriptions are vague and unclear: for instance, cf line 165-167; 169-170, line 237-238 – why partially? Please clarify -; line 245-246 – describe mechanisms if possible/available - ; line 252-253 – please expand, what miRNAs? - ; line 270-272 – please describe markers -; line 273-274 – clarify link between overexpression of caveolin-1 and EMT induction - ; line 312-313 – what’s the source of CAFs in the study? – (list is not exhaustive so please revise manuscript carefully).

As suggested by the reviewer, we have added details for a better understanding. At first, these details were omitted because of the limited number of words.

Line 165-167; 169-170: Descriptions have been developed as requested. (line 203-210)

Line 237-238 : the sentence has been modified as « One mechanism of docetaxel resistance is the transfer of MDR-1/ P-glycoprotein via the EVs [82]». (line 278-279)

Line 245-246 : the mechanism was specified (apoptosis). (line 286)

Line 252-253 : miRNA described were added : « miR3176, -141-3p, -5004-5p, -16-5p, -3915, -488-3p, -23c, -3673 and -3654 regulate AR and PTEN whereas TCF4 are a target of miR-32-5, -141-3p, -606, -381 and -429 » (line 295-296)

Line 270-272  markers have been described : « SNAI1, SNAI2,VIM, CDH2, PAI1 and FN1 » (line 315)

Line 273-274 : for a better understanding, the paragraph has been modified as followed: “Recently, it has been shown that EVs can deliver caveolin-1 into prostate cancer cells leading to cancer stem cell (CSC) phenotype. In LNCaP cells, caveolin-1 induces EMT through activation of NFKB signaling pathway. This change due to EMT induction was associated with increased expression of ZEB1, ZEB2, Slug, Twist, vimentin and downregulated E-cadherin expression. Furthermore, LNCaP cells treated with caveolin-EVs exhibited an increase in migration and invasion capacities as well as higher resistance to docetaxel and radiotherapy” (line 317-323)

Line 312-313:  Sources of CAF have been added when specified in the paper i.e. primary CAF for Zhang’s paper and PSC-27 for Cao’s paper. For Shan’s paper, CAF source was not specified. (line 367-365)

  • The overall writing quality of the present paper is quite poor and need extensive revision. It makes reading complicated at times (cf line 24-25, ‘in prostate field’ line 44…etc). Please revise language / grammar with a native English speaker.

As requested, the paper has been revised by a native English speaker.

  • Line 150-152: a summary of the overall roles of EVs in PCa development and progression (not just resistance) would be a great addition to chapter 2. Try and describe some of the mechanisms mentioned in this paragraph (line 150-152). Also, general introduction to PCa and treatments could be shorter/more concise, so focus is more on the main subject here.

The limited words do not allow us to develop the role of EVs in the development and the progression of PCa. We summarized this part in the section 2: “Accumulating evidence from the last decade publications suggests a significant role of EVs in the hallmarks of cancer [56]. In PCa domain, EVs were shown to be able to induce PCa cell proliferation [57-59], promote the metastasis process through migration and invasion properties acquisition [60-62] and to block the immune system [63]. » (line 185-189)

The general introduction was detailed to introduce the problems of therapeutic resistance.

 The paragraph “Among androgen receptor pathway …… with bone-only metastasis » has been resumed as followed :

« Among androgen receptor pathway inhibitors (ARPIs), two groups can be described: those suppressing androgen biosynthesis (abiraterone) [12] and those blocking AR (enzalutamide and apalutamide) [13]. Abiraterone acetate and enzalutamide have demonstrated significant benefits in terms of OS and PFS either in post-docetaxel setting, chemo-naïve and locally advanced CRPC or mHSPC [14-21]. Concerning Apalutamide, a recent phase III trial led to its approval in mHSPC  [22]. Radionuclides such as radium 223 showed improved survival rates with minor bone marrow damage in mCRPC with bone-only metastasis [23]” (line 70-92)

  • Line 152-155: please remove your own comments from pre-submission drafts…

This paragraph remained from the template. It has been removed.

  • Line 344-346: please clarify that the markers were found expressed in exosomes

The sentence has been modified

  • Table 1 needs to be improved: it’s busy and layout is confusing; different colors / font sizes should be used to clearly discriminate specific sections

The table has been modified

  • Further discussion on the potential of using EVs for therapeutic strategies would be great

We have decided to focus this review on therapeutic resistance in prostate cancer. Several strategies to use EVs as therapy for different cancers are in process with high hope but also limitations. To explore this part of EVs, we need more space, however the review is limited to 5000 words.

  • Try and make conclusion more PCa-specific and expand on current challenges and future potential (clinical) applications

A new paragraph has been added to include clinical trials using EVs in PCa:

“4.3. Future of EVs biomarkers

Several clinical studies using EVs content as biomarkers are conducted nowadays and, more particularly, on PCa diagnosis (exosome Dx; Sentinel™ PCC4 Assay, ClarityDX Prostate) [125]. Concerning therapeutic resistance, very few studies are in process. A clinical study currently investigates the possibile used of AR-V7 for predicting resistance under androgen-receptor signaling inhibitors (PEARL, NCT03601143 ).

If EV content appears a promising source of biomarkers, their characterization and isolation methods need to be improved. Indeed, it remains challenging to isolate EVs from contaminants such as proteins without reducing the quantity of material needed for further analysis. Also, gold standard methods have to be established in order to be applied in clinical context [126]. Furthermore, it remains still difficult to discriminate between different EVs such as microvesicles and exosomes.”  (line 478-489)

Round 2

Reviewer 3 Report

- Writing still needs further improvement. Starting sentences by ‘In PCa’ should be avoided. Also, some sentences don’t seem grammatically correct. See l.22 ‘…cause of cancer death in men worldwide’; figure 1 title; l.53-54 ‘…have been adapted’; l.99 ‘In PCa,…’; l.174 ‘In prostate, production of EVs from normal prostate cells has been described…’; l.206-207 ‘…observed in both androgen-supplied and androgen-deprived (deprived what?); l.250 ‘the most investigated mechanism explaining…’; l.486 ‘in the next future’.

- Shouldn’t the paragraph from l.215 to 221 be before the one starting l.195?

- Introduce the EV acronym in the abstract

- Figure 1: second circle from left, ‘discrimination’ is cut, please adjust font size so the word fits properly in circle

- Table 1 is still not right. It’s also not the best-looking table, efforts to improve presentation would be appreciated. Authors should adjust font size, so words fit better in the different sections (some words are cut). It’s not clear what ‘patients’ means here as there is no indication in the legend. ‘Protein expression’ should be just ‘protein’, so it’s consistent with the ‘miRNA’ and ‘mRNA’ categories. It’s not very clear what the ‘pathways’ category is and what ‘change’ means.

- L.427: the ddPCR and BGS abbreviations should be introduced

Author Response

We would like to thank the reviewer for its new comments and suggestions. As required, we have enclosed a detailed list of the changes, point by point. All changes are highlighted in yellow in the main text for a better readability.

Comments and Suggestions for Authors

- Writing still needs further improvement. Starting sentences by ‘In PCa’ should be avoided. Also, some sentences don’t seem grammatically correct. See l.22 ‘…cause of cancer death in men worldwide’; figure 1 title; l.53-54 ‘…have been adapted’; l.99 ‘In PCa,…’; l.174 ‘In prostate, production of EVs from normal prostate cells has been described…’; l.206-207 ‘…observed in both androgen-supplied and androgen-deprived (deprived what?); l.250 ‘the most investigated mechanism explaining…’; l.486 ‘in the next future’.

All the sentences suggested to be improved have been corrected. We hope that the writing is more appropriate.

- Shouldn’t the paragraph from l.215 to 221 be before the one starting l.195?

We choose to place this paragraph just before the paragraph of the role of AR-Vs in EVs. From our point of view it is easier to follow in this order. To avoid confusions, the precision « AR-V7 » has been deleted line 197

- Introduce the EV acronym in the abstract

The acronym has been added

- Figure 1: second circle from left, ‘discrimination’ is cut, please adjust font size so the word fits properly in circle

The Figure 1 has been modified

- Table 1 is still not right. It’s also not the best-looking table, efforts to improve presentation would be appreciated. Authors should adjust font size, so words fit better in the different sections (some words are cut). It’s not clear what ‘patients’ means here as there is no indication in the legend. ‘Protein expression’ should be just ‘protein’, so it’s consistent with the ‘miRNA’ and ‘mRNA’ categories. It’s not very clear what the ‘pathways’ category is and what ‘change’ means.

We change the table as suggested by the reviewer and hope it is best-looking now.

- L.427: the ddPCR and BGS abbreviations should be introduced

The abbreviations have been added